# Quantum Electrodynamics Coherence and Hormesis: Foundations of Quantum Biology

**DOI:** 10.3390/ijms241814003

**Published:** 2023-09-12

**Authors:** Pierre Madl, Paolo Renati

**Affiliations:** 1Department of Biosciences & Medical Biology, University of Salzburg, A-5020 Salzburg, Austria; 2Prototyping Unit, Edge-Institute, ER-System Mechatronics, A-5440 Golling, Austria; 3The World Water Community, Marconistraat 16, 3029 AK Rotterdam, The Netherlands; paolo.renati@gmail.com

**Keywords:** quantum field theory, phase, coherence, water, symmetry-breaking, dynamical order, resonance, non-thermal effects, hormesis, Weber–Fechner law, Devyatkov law

## Abstract

Background: “Quantum biology” (QB) is a promising theoretical approach addressing questions about how living systems are able to unfold dynamics that cannot be solved on a chemical basis or seem to violate some fundamental laws (e.g., thermodynamic yield, morphogenesis, adaptation, autopoiesis, memory, teleology, biosemiotics). Current “quantum” approaches in biology are still very basic and “corpuscular”, as these rely on a semi-classical and approximated view. We review important considerations of theory and experiments of the recent past in the field of condensed matter, water, physics of living systems, and biochemistry to join them by creating a consistent picture applicable for life sciences. Within quantum field theory (QFT), the field (also in the matter field) has the primacy whereby the particle, or “quantum”, is a derivative of it. The phase of the oscillation and not the number of quanta is the most important observable of the system. Thermodynamics of open systems, symmetry breaking, fractals, and quantum electrodynamics (QED) provide a consistent picture of condensed matter, liquid water, and living matter. Coherence, resonance-driven biochemistry, and ion cyclotron resonance (Liboff–Zhadin effect) emerge as crucial hormetic phenomena. We offer a paradigmatic approach when dealing with living systems in order to enrich and ultimately better understand the implications of current research activities in the field of life sciences.

## 1. Introduction

One of the main reasons why living matter challenges the descriptive capacity of the sciences so much is due to the fact that dealing with living systems essentially means dealing with open processes, flows, and relationships. Living systems are never isolated, are self-contained, and thus elude us as observable objects. A living system is not a finite “body” but is always in flux, existing as a process, as “ordered responses” along a thermodynamic history of couplings and adaptations. This fact is difficult to “handle” with the prevailing scientific jargon, that is, when trying to reduce a living entity to the physical realm of how and why living dynamics emerge. We must come to terms with the fact that the language of the physical sciences and the commonly accepted condensed matter view [1] is only appropriate for describing objects and observables that exist in themselves. In fact, this is not the case for living systems in two respects:(i)Living things exist as transduction devices of stimuli into responses, as transformers of inputs and outputs, according to deterministic, non-random, but not predeterminable causal relations [2];(ii)The theoretical tools commonly used to analyze the dynamics of living things are not capable of exhaustively describing the typical features of living things (from biochemical orders to energetic yields and biocommunication to morphogenesis, adaptation, non-random evolutionary pathways, as well as memory, behavior, choice, and teleology) [3,4,5].

In this paper, we present relevant considerations and perspectives on these two crucial issues by showing why living systems must be described by a theoretical framework that admits many vacuum levels and symmetry breaking as a conditio sine qua non (i.e., without which no kind of dynamical order could ever emerge). 

## 2. Shifting from QM to QFT

### 2.1. Order, Symmetry Breaking, Dissipation, and Coherence

Today, “quantum” approaches are becoming more common in biology [6], simply because some properties and phenomena cannot be described classically. However, most of these approaches still rely on the semi-classical theoretical framework of quantum mechanics (QM), which is not only an incomplete theory [7,8] but is also unable to describe the phase transitions that constitute living matter. 

Biological matter has the special property of being composed of long-range correlated components. One might be tempted to think that the components (types of matter quanta such as molecules, ions, electric charges, atoms, and electromagnetic fields) in soft matter (alive) are identical to the inanimate state (not alive) and that “something mystical” causes them to function in a “special way” to yield the alive state. As will be shown, this vitalist-tinged perspective suffers from a postulate that holds only if one remains faithful to the classical or semi-classical view of matter (or what we consider “physical events”), where the ground level (minimum energy) of a system (even of a single atom) is uniquely determinable (for classical physics, CP) or uncertain (within the Planck limit) but nonetheless unambiguous (for quantum mechanics, QM) [9,10]. In the conceptual framework of QM and in accordance with the fundamental theorem of von Neumann [11], a unique ground state (vacuum) of the ensemble of interacting molecules interacting by forces does indeed exist. Therefore, a unique state (or phase) is admitted whose phase transitions are not describable [10]. The property that characterizes a living state and gives it its dissipative character has already been exemplified and is accessible in numerous articles [12,13,14,15,16].

Since a biological system is constantly connected to the external environment (open), its isolation means the elimination of its functionality, its destruction that ultimately will be lethal. As addressed by Vitiello [3,17], when studying an open system (say, “system *α*”), we proceed to “close” it, also considering the environment in which it is immersed. We can refer to the latter as “system *β*”. Doing so assures that the flow equilibria of matter, energy, and phase correlations between the system *α* and the environment is still describable through thermodynamics of closed systems. With respect to energy exchange, the energy output of *α*, E(*α*), must be equal to the energy input of *β*, E(*β*), and vice versa. In either case, it must be E(*α*) − E(*β*) = 0. Therefore, the set (*α*, *β*) of systems *α* and *β* behaves like a “closed” system for which there is no energy flow, neither input nor output.

Given the energy balance (and the balance of flows of all other quantities exchanged between *α* and *β*), system *β* behaves like a copy of system *α*, in the sense that it behaves the same as system *α* in terms of flows, provided they are reversed: what is an input for *α* is an output for *β* and vice versa. Thus, reversing the direction of flow is equivalent to exchanging *α* with *β* or vice versa. Since technically the flow direction is reversed by changing the sign of the time variable, we can say that *β* behaves like the copy of *α* for which the time arrow has been reversed (*β* is the time-reversed copy of *α*). In essence, *β* is the system that describes the environment in terms of the balance of *α*’s energy flow, while it is also the mirror image of *α* in the mirror of time (*β* is the time mirror image of *α*): Vitiello effectively expressed this fact by writing that *β* is the “double” of *α* [17]. Thus, in the case of open systems (such as organisms or brains), we need to consider their doubles, and this offers some interesting insights. The formation of each configuration (*α*, *β*) is enabled by the breaking of symmetry induced by the external stimuli, so that:(i)The manifold of possible configurations is made possible by the existence of the numerous (infinite) possible ground states in the scenario offered by QFT;(ii)The coexistence of these multiple configurations is given by the fact that these ground states are orthogonal to each other;(iii)Their temporal sequence is given by the dissipative dynamics, i.e., by their thermodynamic history (by all possible pairs of values of N(*α*) and N(*β*) obeying the relations N(*α*) − N(*β*) = 0);(iv)The succession of the states of the system is indeed a time-dependent thermodynamical (dissipative) history along which the following states depends on the previous ones in a deterministic, but *a priori* unpredictable, way because of the presence of its thermodynamical “double” (environment, inclusive of every possible quality and kind of stimuli).

Based on the theoretical foundations of quantum field theory (QFT), the structural and functional order of a biological system (which is neither stochastic nor predeterminable) results in a condition that arises dynamically as a consequence of symmetry breaking over a wide range of degrees of freedom of the system [10,18].

In spontaneous symmetry breaking (SSB), the symmetric properties of the microscopic dynamics imply the presence of fields which, distributed throughout the system, cause the formation of ordered configurations of the elementary components. The wave character of these fields is associated with quanta, whose role is that of a messenger or “carrier” responsible for the correlation and order between the quanta of matter and the transmission of their quantum state to others. 

The quanta associated with such dynamics (i.e., SSB) are called Nambu-Goldstone (NG) quanta (or particles or modes) [3,19,20,21], and include the phonons in crystals, the magnons in ferromagnets, the polarons in ferroelectrics, etc. Quanta of NG and responsible for the space–time arrangements observed in such systems, which, depending on the case (geometric, magnetic, rotational, electric, etc.), carry the specific correlations that generate the observed order. Even at very different scales, as in cosmology [9,22], or where non-trivial forms of order occur, as in biological systems (a very special case of condensed matter), the breaking of several kinds of symmetry underlies the appearance of order parameters [3,10,19,23,24]. 

The NG quanta are bosons, i.e., many of them can occupy the same physical state [25]. Under ideal conditions (without “edge effects”) they have zero mass, and in their low-momentum state they do not contribute to the energy of the ground state (vacuum): the emergence of order is therefore a condensation effect of the NG bosons in the vacuum (ground state). The NG quanta are the mediators of correlation and are structural elements. They are real elementary components, which refer to the dynamics of the systems (not separable from the structure itself) and can be measured by scattering techniques and from which a spectrum can be defined [26]. 

To illustrate that order is the dynamical consequence of symmetry breaking (in which correlation bosons condense between the components of the system), let us begin with a simple example: the formation of a crystal lattice by the cooling of a liquid or vapor.

In a liquid or gas, the atoms (or molecules) are in a state where their positions in space can be shifted without changing the macrostate. A shift in any direction of space therefore produces a configuration equivalent with identical free energy (and entropic content). In practice, a gas or liquid are symmetric systems with respect to the degrees of freedom of spatial displacement. For example, when the system is cooled below a critical threshold (the condensation temperature), this spatial translational symmetry is broken, so that atoms can only occupy certain places at certain reciprocal distances (i.e., integer multiples of the lattice constant). In a field perspective of condensed matter, this condition is no longer restricted to short-range forces (see the criticism in [1,27]) but gives rise to long-range correlations in the form of stationary elastic waves such as phonons. These virtual acoustic oscillations, when compressed, slip into a common phase to reach a minimum energy level (vacuum) lower than the one they had in the isolated state (vapor phase) or less correlated (liquid phase). 

Therefore, when the system is open and capable of dissipating a certain amount (i.e., entropy, latent heat of condensation), a phase transition from a disordered state to a more ordered state occurs spontaneously [1,10]. This new state, the vacuum level, is energetically lower than the previous one. This energy gap (in electron volts, eV) characterizes its thermodynamic stability. The translational symmetry is broken, and it is no longer possible to move along another direction of the system experiencing the same potential landscapes. 

This suggests that (i) continuous symmetry breaking as a result of the guided pathways in biotic systems that seeks to maintain a viable homeostasis leads to a lack of symmetry and (ii) is the consequence of quantum dynamics in which the components of the matter field phase their oscillations and create a correlation that manifests itself as a field consisting of a discrete set of bosons (in this case phonons) [17]. This correlation field is indeed a classical field (the order parameter), since it is insensitive to quantum fluctuations and causes the macroscopic (classical) stability of large quantum systems (such as a crystal) [10]. 

However, phonons cannot be “detached” from the system, i.e., there are no freely propagating phonons outside a crystal because they exist only as long as the crystal exists. This implies that the quanta of NG are identified with the function of the specific order for which they are responsible and thus also express the functionality, the mode of being of the system. Without them, the system is “another system” with completely different physical properties (functionality and therefore structure). 

Since the NG bosons determine the phase correlation over large distances of the elementary components, a change in the degree of condensation is equivalent to a change in the intensity of the correlation over large distances. This correlation is called coherence, and the condensed state is called the coherent state (over a certain degree of freedom). [28,29]. The stability of the condensed state referred to is thus the (thermodynamic) stability of the coherent state (see Appendix A for a brief overview of the non-coherent thermodynamic approach). 

As we can see, the QFT approach resolves the Cartesian dualism (from the earlier Platonic and then Aristotelian heritage) between structure and function, which, depending on the context, takes on many versions (e.g., form–substance, information–matter, software–hardware, psyche–soma, soul–body, mind–brain, God–cosmos) [30]. 

The formalism of QFT allows a unified view, as it describes a variety of non-trivial phenomena through the dynamical relation between the microscopic and the macroscopic (mesoscopic) level, without the need for ad hoc additional operations. Essentially, these bosonic condensates make a scale-free transformation possible (from the micro to the macro level), with its macroscopic behavior being not “superimposed” but rooted in the quantum description of the microscopic components. Such a scale-free system is termed a “macroscopic quantum system“ [17], since its macroscopic properties are understandable only in terms of the underlying quantum dynamics of the elementary components. 

Overcoming the Cartesian dualism makes it necessary to allow the existence of a variety of vacuum levels. This is only possible by referring to QFT, where phase transitions and symmetry breaking can be described dynamically. Such a descriptive power is not practicable in a classical or semi-classical theoretical framework, such as QM [7]—which admits only a single ground state (thus no phase transitions) [10]. 

### 2.2. Condensed Matter from a Field Perspective: Water, the Matrix of Life

The principles of SSB described above—applied to condensed matter such as liquid water—allow a fully consistent picture to emerge accounting for phenomena occurring in both plain water and living matter. 

Once a critical threshold density (below a critical temperature) is reached, SSB induces condensation of matter in coherently coupled domains [10]. For water, such thresholds are reached at 0.32 g/cm^3^ and 373.15 K (at 1 atmosphere pressure) [31]. 

Experimental evidence [32,33,34,35] and descriptive approaches [36] have demonstrated that liquid water must be a two-phase system. According to QED [1], part of the molecules in such a system (whose abundance is in “inverse proportion” to the temperature) are organized in collective, coherent vibrational domains corresponding approximately to the size of a wavelength of the coupled electromagnetic mode. Their energy range corresponds to the spectral distance between the two energy levels at which the electron of each water molecule oscillates. Such regions are called coherence domains (CDs), and, in liquid water, where coherence is established between the sp^3^ and 5d levels, their size is ideally the wavelength of the corresponding electromagnetic mode (photon), about 100 nm (neglecting thermal fluctuations, which correspond to a frequency of the field in vacuum of 3 PHz, which, during the coupling with the matter field given by the resonating molecules, will be renormalized down to the IR range, about 6 × 10^13^ Hz). At non-zero Kelvin temperature, its size is reduced by thermal noise, which detunes some of the molecules, generating an incoherent vaporous “halo” around the coherent domain [31,37]. 

The new ground state (known as a “spontaneous superradiant phase transition”) associated with coherently oscillating molecules is lower than that of incoherent, isolated molecules. The difference of this energy gap expresses the thermodynamic stability of the coherent state with respect to decohesion agents acting from outside (such as temperature, photons, fields in general, and mechanical forces). If the excitations are small enough (smaller than the energy gap), the CD receives them as a whole; if they are larger, one or more oscillators are released from coherence and “spilled back” into the normal (incoherent) part [1,37]. For ordinary liquid water—where coherence is based on electron cloud oscillation for each molecule—this energy gap is of the order of 0.2 eV (depending on temperature and position within the CD—usually smaller at the periphery rather than at the center) [31]. 

If the fluid is confined by hydrophilic surfaces, the coherent fraction is stabilized, and possibly other degrees of coherence (such as dipole rotation) can be set. This is the typical condition in living matter as described further below [38,39]. 

As water CDs are self-generated total-reflectance cavities for the EM field, the presence of an externally supplied cavity and pump (as is the case in laser physics) is not required. Indeed, the self-trapped electromagnetic field cannot be externally irradiated as it an intrinsic phenomenon related to the coupling of molecules and as such implies total internal reflection [37,40]. Interestingly, almost four decades ago, Persinger [41] noted that the effects of therapeutically active substances are due to their electromagnetic oscillations rather than due to the biochemical properties; that is, a drug’s electromagnetic oscillations already induce an effect before it is ingested. 

Each molecule participating in the coherent dynamics is endowed with an electron that spends about 10% of its time at the 5d energy level, 12.07 eV above the sp^3^ hybrid orbital (ignoring collective storable electron excitations). This level is at most 0.53 eV below the ionization threshold (I_th_), which for water is 12.60 eV. Thus, we can say that there are 0.10 quasi-free electrons per coherent molecule. In a CD there are about seven million molecules, yielding about 0.6 ÷ 0.9 × 10^6^ quasi-free electrons per CD. 

These quasi-free electrons can also escape from a CD through the quantum tunneling and in doing so become available to suitable reaction partners (such as the oxygen molecule, non-coherent water molecules, or other species) or are extracted by small excitations [42]. Tunneling of electrons from the CD leaves behind ionized water molecules. These ions no longer participate in the coherent oscillation and therefore pass to the non-coherent fraction—exactly as described in the Landau model of liquid helium [37]. 

The extracted electron on the other hand could be captured by an oxygen molecule dissolved in water, creating a negative ion. The resulting ion pair, after some chemical steps [43], eventually leads to the formation of a proton and reactive oxygen species such as a hydroxyl. Thus, the CD is an electron donor and then a chemical reducer, which, together with the non-coherent moiety, forms a “redox pile” [4,43,44]. In short, CDs are a prime driver of the branching chain reaction (BCR) of water [45,46]. A remarkable example for this reaction mechanism is an experiment (Figure 1) where a water sample is stimulated by a NIR laser to a pump reaction which leads to an electromagnetic radiation in the blue-light range [47]. 

Because water CDs are easily excitable, they can collect many small external excitations (high-entropy energy) and generate unique coherent vortices whose energy is the sum of all small, collected energy excitations but whose entropy is low (high-degree, low-entropy energy) [48]. However, this collective energy cannot be thermally released to the outside, which explains the long lifetime of the coherent excited states within the domains—hence the name “cold vortices” —and also in living matter, as suggested by A. Szent-Gyorgyi decades ago [49,50]. 

In turn, to generate coherence between the CDs, it is necessary to make them oscillate, which means that in one part of the oscillatory cycle the CDs should be able to release energy to the outside, and in the other part of the oscillatory cycle they should recover this energy from external excitations [4,16,48]. 

One way to do this is to release energy quanta to molecular excitable species that are able to align some of their vibrational modes (energetically excited states) with the energy released by the water domains. In this case, chemical utilization of the obtained energy could occur, since the excited (activated) molecular species could (more easily) participate as reagents in chemical and electrochemical reactions [4]. 

Thus, if external non-aqueous molecules present in the fluid contain in their own spectrum at least one frequency close to the vibrational frequency of the water CD (resonance), then these molecules could participate in the coherent dynamics of the fluid and would be subject to dynamics that we now outline [42]. 

In similar fashion to the BCR, a hormetic effect employed by ultra-weak electromagnetic field stimulations can be explained. This is especially the case for the sensitivity of migrating birds capable of detecting changes in the average magnetic field valued at the level of 10–100 nT (Figure 2). Resonant coupling of water CDs of biogenic structures within the extremely low-frequency regime well below the thermal threshold k_B_T results in macroscopic effects within biota. Similar effects occur in the frequency range of microwaves, even when the power of EM radiation is far from that at which thermal effects are possible [51,52]. A brief summary of the effects of weak magnetic fields on biota is provided in Appendix B. 

### 2.3. Living Phase of Matter: Time-Dependent Order and Morphogenesis

The molecules belonging to the coherent fraction form a set in which the phase is well defined: their common wave function is defined by eigenstates of the phase (*φ*), complementing the observable “number” (*N*): the uncertainty relation (Δ) expressed in natural units (with *h*/2π = *c* = *k_B_* = 1, where ”*h*” is Planck’s constant, ”*c*” is the speed of light in vacuum, and ”*k_B_*” is Boltzmann’s constant), results in the “fundamental uncertainty relationship” expressed as Δ*φ*Δ*N* ≥ ½ [1]. 

In a perfectly coherent state, the number of oscillators is completely indeterminate, while the phase, the wave-like aspect of the field, is precisely defined. This means that in a coherent state, the individuality of the oscillators loses physical meaning, since the matter field of uncountable quanta coupled to an electromagnetic field whose massless part defines itself as a quasiparticle in the coherent phase [53,54,55,56,57] is the only definable object [1,7]. 

This is equivalent to minimizing the uncertainty of the phase (Δ*φ*) and thus maximizing the uncertainty of the number of quanta (Δ*N*). In order for this to happen, the system tends to have a large number (*N*) of quanta because Δ*N* ≤ *N*, with a continuous transition, i.e., an exchange between the coherent phase and the external environment. Since the coherent state is thermodynamically more stable (lower vacuum level than the disordered state), coherent systems tend to share their oscillations with other systems, with which they can resonate as to increase *N*. This is a fundamental feature for understanding that living systems are de facto (super)coherent systems. The open flow of matter and energy quanta is thus possible as they share a common vibrational phase with everything capable of doing so. 

Since phase correlations are nonlocal correlations that do not imply an exchange of kinetic energy [58,59], the definition of the term “ecosystem” in the theoretical framework of QED is appropriate. In this context, the term “ecosystem” is considered a region of spacetime, in which living systems share an oscillation phase of certain electromagnetic modes that cover very large spatial domains (extending over kilometers and more) at long wavelengths [15,60]. 

Due to the interplay with other molecular and ionic species, as well as the ubiquitous presence of interfaces, coherent water domains in biological matter are further stabilized [61,62]. Compared to an ordinary liquid, this fraction is <50% at room temperature and pressure (Figure 3) [63]. In biota, the degrees of coherence are not just numerous and interconnected but also mechanically constrained [4,48,64]. As a result, biological matter, with *N* being >90% water, is in a sol–gel state and in a sense “solid” [65,66,67]. 

Quantum-mechanically, the density of water does not significantly alter with or without a solute. When the solute concentration is high, the concentration of water coherence domains is low. As the concentration of the solute decreases, the concentration of water coherence domains increases [67] Thus, by modulating the ionic concentration, the cell is capable of regulating the degree of water CDs. In fact, this gel-like state is of vital importance in rapid protein deactivation (Figure 4) as well as the transport functionality of transmembrane protein complexes such as aquaporins. Given that the water molecule is larger than the diameter of the aquaporin (e.g., AQP1), the water molecules must be present in a superposition of states on either side of the channel to “tunnel” through it. This implies that, given the small size of the water molecule, it no longer behaves like a particle but as a wave [67]. 

The size of coherent domains is never smaller than the maximum distance between any surface (membrane, molecular chain, etc.) and another [42]. Such supramolecular organization of water molecules into CDs has two major important implications: (i) the topology of the quasi-free electrons circulating around CDs cannot be tridimensional [69,70] (this does not exclude that CDs occupy 3D volumes and can exist in bulk pure water [32], in contrast to Sen et al. [69]) and (ii) no supercoherence can be observed in pure liquid water [42,71]. In order to self-organize on further levels, CDs require the presence of other species with which to resonate or some surfaces such as nano-particles (e.g., ions or bubbles) and/or hydrophilic walls [4,67,72]. Only then will the entire water matrix in living matter be able to experience multimodal coherence, termed supercoherence, which allows for general “multiplexed” phase correlation throughout the system—be it a single cell or a multicellular organism [4,62]. In biological systems, many degrees of freedom over the components are engaged in coherent dynamics, and many kinds of symmetry are thus broken (known but not yet catalogued, among which we can guess: oscillations of many molecular species, ions and electrolytes, membranes, enzyme and protein folding and unfolding, microtubule pulsations, etc.; however, a whole phase correlation over the ubiquitous water matrix is in force, like in a jazz ensemble, as, indeed, referred to by Mae Wan Ho [73,74,75], who compared biological coherence to “quantum jazz”). As supercoherence is the result of interacting virtual photons that tie together individual CDs, it follows that water must be highly sensitive to electromagnetic fields—especially in the ultraviolet and far-infrared [67]. Similarly, electrons trapped within the coherent plasma make CDs behave as small magnets. This in turn makes water highly sensitive to magnetic fields and radio waves as well [67]—an effect previously documented in the phenomenon of burning salt water upon radio-wave exposure [76]. 

One of the most difficult aspects of molecular biology to understand has always been the fact that biochemical processes occur with incredible efficiency, with an extraordinary degree of precision and timing [43,64]. This capacity for biochemical activity also allows living matter to express one of its most challenging properties that cannot be simulated in vitro, namely the ability to perform chemical reaction cycles [64] that are perfectly selected from a large number of educts that are present together in the cellular landscape—a good example is the Krebs or Calvin cycle. How can this be explained? 

In trying to summarize this very illustrative aspect (for details, see [2,4] it should be remembered that any chemical species is first and foremost a physical species, i.e., an oscillator characterized by its own precise proper frequencies (modes). Since coherence based on a particular mode of vibration is a dynamic in which only oscillators capable of resonating on that mode participate, it is clear that the chemical species involved in the assembly of biological matter are those that possess oscillation modes equal or very similar to those of the supercoherent aqueous matrix. 

In the coherent phase, there are regions where a background field oscillation acts as a director and coordinator of the molecular encounters. Inside the CD, however, and due to the high dielectric constant $\varepsilon_{coh}\sim 160$, fields are strongly shielded (see Appendix C for the theoretical description). The algebraic relations presented in the Section C.1 describe the reason why there are ordered and efficient reaction pathways in living matter. Namely, the first reaction expresses how each set of electric charges arriving at the periphery of a CD is strongly polarized by shifting the lighter charges (typically electrons) much more outward than the heavy charges (nuclei): this polarization leads to a strong instability of the molecules, which makes them react much more easily and accelerates the reaction kinetics by lowering the activation threshold [4]. The other two equations express that only certain species non-randomly approach the surface of the CD: at a given time, only molecules with the appropriate natural frequency (resonance frequency) can encounter each other (typically at the CD interface, where the field gradient is largest). 

Once the reactive species have been brought together and activated, the water CD catalyzes the biochemical reaction (typically an oxidation–reduction reaction) by releasing electrons (which in the coherent phase occupy states close to the ionization threshold and are therefore readily released at charges below 0.4 eV [72] or by emitting field quanta I the form of biophotons [61,77,78,79]. However, this mediation implies a change in the state of the CD with a consequent change in its own frequency, whereby this CD is now capable of attracting other reagents, thus enabling a second reaction step, forming a cascade to give birth to the abovementioned Krebs or Calvin cycle. 

These reaction paths (guided by the sequence of vibrational modes of the coherent fraction) are deterministic and by no means random. On the other hand, they are also unpredictable, since they are determined by (i) the preceding overall thermodynamic history at a given time, (ii) the coherent configuration of the CD at that time, and (iii) the current boundary conditions according to the available reagents and their concentrations, ambient fields, phase correlations, macroscopic thermodynamic variables, etc. These reaction paths influence the modulation of the natural frequencies of the CDs and thus the way in which a given input is converted into an output. 

Supercoherence of living matter consists of the establishment of further levels of coherence (coherence of CDs) due to the dialectic of coherent water with other molecular species that act as receivers of quanta of free energy released by the CDs—mostly in the form of electrons or photons or rotational excitations where angular momentum transfer occurs [1,31]. This implies that these must not thermally relax, otherwise they would lose their coherence at the expense of energy [31,37]. In doing so, CDs act in concert with other biomolecules as multimodal laser devices that extend coherence to the next hierarchical level, from which others and further emanate in a retroactive and dialectical genesis [42,64,72]. 

Supercoherence is a crucial condition for the emergence of stimulus–response laws. Any stimulus provided, being “weak” such that coherence is not destroyed in its entirety, is received by the totality of the coherent whole. This state also reflects the fractal principle as it is in essence a scaled-up version of the constituting smaller entities (CDs)—here in particular, the electromagnetic sensitivity has been assigned to DNA [80] as well as protein folding patterns such as those found in α-helices [81]. At hierarchical levels, fractal properties can be found in any bifurcating structure—be it plantae or in animalia—and are characterized by a huge gain in electromagnetic sensitivity [82]. On a macroscopic level, single-molecule events affect the whole organism, which is subject to the concerted supercoherent state, leading to the emergence of a self. This is possible as each oscillator species shares, on at least one degree of freedom, the phase eigenstate with others, which in turn share it on yet more degrees, and so on, to ultimately establish a network of interconnected and interdependent coherences [4]. A practical example illustrates this suitably. A nutrient perceived by an amoeba implies that the possible interaction between receptor and ligand is actually an event “known” to the whole amoeba. Since this receptor is part of a chorus of oscillators sharing a phase eigenstate, the state change of “one of them” is the state change of the whole so as to inform all subunits of the amoeba. The supercoherent matter field with its well-defined phase has become a physical reality [1,10,83]. In accordance with Batson’s famous quote of “Information is a difference that makes a difference”, this is a first step that determines the difference between “measuring” the composition of a food source and “experiencing” its taste [2]. 

### 2.4. A Sensible Case: Ions and Ultraweak Fields

Although dissolving salt in water is a common and everyday occurrence, it holds very deep mysteries: How can solvation begin? How can a small molecule like water, which is about 3 Å in size and whose dipole amplitude is no more than 0.38 Å, be able to detach ions from an ionic crystal whose stability is determined by a considerable bond strength (bond strength for most common salts is approx. 5 eV)? Although it is known that the dielectric constant of water at room temperature is about 80, the question arises of why calculations based on the molecular properties of H_2_O using the Langevin equation give a value of about 13. And why does the solvent power increase with an increase in temperature while at the same time the dielectric constant decreases? These questions cannot be answered consistently from either the CP or the QM points of view and can only be explained by considering coherence in liquid water. The empirically measured high value of the dielectric constant in liquid water (*ε* = 80 at 300 K) applies only macroscopically to the bulk liquid. This could justify the steady state of the already dissolved ions, which provides a rationale for their separation from each other. But it does not realistically support any process involving the onset of dissolution (its dynamics). 

The first transition of the electric dipole of water occurs at about 7.4 eV, i.e., in the full UV spectrum, and the oscillator strength (f) of such dipolar transitions is very low (f∼0.05) [84]. Thus, the water molecule is an electric object that is rather rigid, which contradicts the high polarizability required by the pair potentials used in the second-order perturbation theory. This means that the only electrostatic effects water can produce are related to its modest electric dipole and are too tiny to keep oppositely charged ions apart and separated. The attractive Coulomb potential between charged ions in the crystal lattice of the salt can be overcome only if a lower-energy state is accessible; a component is needed that must be able to balance the large enthalpy of ionic bonding, which reaches values in the range of 5–7 eV on average in common salts. The dissolution of ions can take place only if the dielectric constant of water is lower than the known value of *ε_bulk_* = 80 (in fact, within the QED picture, the incoherent phase has a value of *ε_inc_* = 13, whereas the coherent phase of water with *ε_coh_*~160 is quite high (see Section C.2 for details). It can be concluded that the ions oscillate in phase with each other and in a cross distribution of ions and counterions. Thus, the energetic gain of the system Δ, which allows it to overcome the strong stability given by the Coulomb ionic bonds, is associated with the creation of new dynamical structures that are ordered and coherent and embedded in the noncoherent phase of the solvent.

As we see, the energy gap depends only on the temperature and the value of the dielectric constant of the incoherent part of the liquid (which in turn also depends on the temperature), and this energy gap corresponds to values of the order of a few eV. Each ion coordinates around itself a 3D plasmatic cage of counterions (with opposite charge). The constraint of coherence (i.e., all ions must vibrate in the same phase and at the same frequency as their counterparts) means that the counterion cages must be identical to avoid collisions (collision implies that these cages to be torn apart), with the concomitant loss of coherence. This comes at an energetic cost and therefore does not occur within certain concentration and temperature thresholds. Furthermore, once temperature increases, the solubility in water increases with the rise in the incoherent fraction of water, increasing the space available for ion dissolution (Figure 5). 

On top of that, coherence prevents ions from being subjected to thermal collisions: it is clear that ion ensembles are noiseless inside. The only noise is that coming from the collision of the collective plasmas (cages) with the incoherent molecules of the solvent. Such collisions do not produce any recoil in the ions, since they respond as a whole and all move in unison. 

The fact that the ions cannot be subjected to collisions, as presented in the common “Brownian picture”, is confirmed by the fact that, if it were true, a trivial solution of NaCl in water would emit radiation (bremsstrahlung) as ions are accelerated charged bodies. If one assumes that each collision releases an energy of about 10^−17^ eV, the solution should release about 10^−5^ eV/s per ion, and a standard electrolytic solution at room temperature should release all of its energy (and freeze in the process) within 42 min [85]. As we all know from daily experience, this is not the case. 

A very striking phenomenon definitely refutes the picture that imagines ions as isolated (or singular) dissolved charges subjected to random motions and collisions in the gaseous incoherent fraction of water. It is the Zhadin effect [86,87,88,89], which is nothing but an ion cyclotronic resonance process [85,90,91]. 

Liboff [90] and Zhadin [86] have independently shown that in an electrolytic cell in which nucleic acid (glutamic acid) is dissolved in water, an electric current spike occurs after application of a weak electric voltage in the presence of a parallel weak static (*B_dc_*) and a very weak dynamic (*B_ac_*) aligned magnetic field. This occurs due to the interplay of the static (few tenths of a μT) and the alternating (nT range—according to [87], 20 nT are sufficient) field, with the later oscillating at a frequency corresponding to the cyclotron frequency of the ionic solute species (Figure 6). Because these fields are weak, they do not provide sufficient energy to the solution for the ionic current to overcome the thermal noise to produce the peak shown. This phenomenon subsides once the frequency of the alternating B-field falls outside the resonance value or when the amplitude of both fields exceeds their corresponding threshold values. The frequency and intensity window (Adey’s window) is therefore narrow and specific to each ionic species [92,93]. 

If the resonating ions would obey Brownian motion, they would travel short distances between collisions with rather high speed (several 100 m/s), generating collision rates of the order of trillions per second. In fact, the effective drift velocity generated would be too small and incompatible with the experimental observation (yielding no electric current spike). 

The values of the radii of the trajectories that the ions should traverse in the classical picture pose a problem because they are in the order of 0.4 ÷ 2 m—a reason why many scientists have rejected that the Zhadin effect could be due to ion cyclotron resonance. However, this problem can be quickly resolved by considering that the actual velocities at which ions move are much lower than predicted by the Brownian image. To abandon the Brownian picture, we must move to a QED perspective, both for the solution medium (liquid water) and for the dissolved charges (ions). 

This phenomenon can be quantitatively explained when returning to the above concept of the coherent plasma in the non-coherent part of the solvent. Doing so requires two preconditions:-The “interface” zone, where self-trapped EM fields decay from inside water CDs, is the region between the ideal radius (*R_CD_*) of a CD at zero temperature and its actual radius due to thermal erosion from outside incoherent molecules. This zone has a temperature-dependent thickness *δ*. The effective radius r of a CD is thus given as *r = R_CD_* − *δ(T)*. At normal temperatures (20 °C), *δ*(*T*)~10 Å.-The “reservoir” zone corresponds to the interstices in-between CDs. Such zones are not occupied by EM field tails, and water is dense and vaporous there (Figure 7).

In both zones, the ions produce coherent plasmas (“non-ideal Debye–Hückel plasmas”) in the “reservoir” zone. The ions in the “reservoir” zones do not produce cyclotronic motion and do not contribute to the transient current spikes that occur when resonating with external fields. The only ions that generate a (non-dissipative) electric current for cyclotronic resonance with a (weak) B field are those located in the “interface” zone, where the EM field decays. 

Thus, the current spike generated in the Zhadin effect relates to the emptying of the “interface” zones, with a migration of their ions towards the electrodes (see Section C.3). Practically, this implies that only ions aligned to the resonance frequency of the applied alternating magnetic field are driven out of orbit and meander in an almost frictionless state through the interstices of the CD lattice. The resulting ion current is depicted in Figure 8.

### 2.5. Hormesis: The Stimulus–Response Relationship

Hormesis (a creation from ancient Greek ὁρμέειν (*hormáein*), meaning “to set in motion, drive, urge”—a term coined by Southam and Ehrlich [94]—describes a favorable biological response to low doses of toxins and other stressors, which activate an adaptive stress response that raise the resistance of the organism against high doses of the same agent [95]. Thus, hormesis refers to the stimulatory, non-linear effects at low concentrations of otherwise toxic substances. In biological systems, any physical, chemical, or biological agent can be considered stimulatory when administered in much lower concentrations than when administered in undiluted, harmful concentrations [96]. While hormetic effects of ionizing radiation have been known for quite some time [97,98], only recently has the biomedical community started to exploit hormesis for public health, therapeutic, and commercial benefits [99]. 

Whether hormesis is a “generalizable” concept valid throughout the animated world is the subject of ongoing discussions. So far, this process addresses (1) evolutionary aspects, (2) how widespread it is among biota, (3) where it is integrated in physiological response systems, and (4) how environmental physicochemical stressors trigger it [100]. While an extensive literature study showed that 20–65% of the chemical agents investigated display biphasic response patterns [100], the corresponding observations of a purely physical agent such as non-ionizing radiation are still regarded as non-existent. Thus, in accordance with the proposed harmonization of the numerous expressions describing the biphasic dose–response pattern (see [101]), it would be suitable to group electromagnetically induced hormesis within the category of “radiation hormesis”, even though the radiation involved is non-ionizing. Even if this extension is considered, however, a distinct difference must be taken into account, which differs from radiation hormesis. As further elaborated in Appendix A, at wavelengths shorter than the IR window (the Wien region), a logarithmic dependence is observed for which the Weber–Fechner law (WFL) is most appropriate [102]. At wavelengths longer than the IR window, denoting the Rayleigh–Jeans region, the situation is quite different, as there is no gradual dose–response pattern. Instead, one is confronted with a sudden “step response” as a given threshold is reached. Although a biphasic response pattern seems to exist [103,104], the sudden response independent of a gradual increase in stimulus intensity prompted Chukova [105] to speak of the Devyatkov law (DL), rather than the WFL. Although WFL has its origins in psychophysics, certain physiological responses of electro-perception point directly to electromagnetic susceptibility, most notably Lorenzini’s ampullae of cartilaginous fish (sharks and rays), revealing perceivable voltage gradients as low as 1 μV/cm [106]. In fact, the author concludes that the receptor mechanisms involve such high sensitivity that it is comparable with the single-photon or single-molecule threshold sensitivity of rods (eyes) or antennal chemoreceptors (of insects).

As outlined above, ultra-weak fields are paramount as these provide the basis for the non-dissipative flux of electrically charged metabolites (mostly ions) within coherent systems. Cyclotronic ion resonance is further evidence for the applicability of both the WFL and DL as these are consistent with the characteristics of coherent systems for the following reason: if the energy input is too high (overcoming the energy gap), the system responds with a loss of coherence of one or more components. In this case, the system does not update itself with respect to the stimulus. Only if the energy input is small enough (minimal stimulus) can the system respond coherently and collectively and update its eigenstate in accordance with the received stimulus. The relevance for medicine in this respect becomes immediately obvious: intervention from outside when a ‘therapeutic stimulus’ should ‘update’ the organism without endangering or discarding its own stability. Any stimulus that exceeds a given intensity threshold becomes useless (or even harmful) in as much as weakening the stability of the system in the long run.

The word “law” in this context indicates that it has proven to be a good fit to experimental data in physiology and psychology [107,108] and in botany [109]. The WBL states that the physiological or psychological response of an organism is a linear function of the logarithm of the stimulus magnitude. Mathematically, the WBL can be written as $R\;\sim \;log\left(S\right)+M$, where *R* is the response (e.g., the number of nerve impulses emitted by a sensory organ over time), *S* is the stimulus magnitude (e.g., the intensity of light to which an eye is exposed), and M being a constant. The frequent occurrence of the WBL suggests that it may be a fundamental physical or chemical property of soft matter. Tosi and Del Giudice [110] have demonstrated that the WBL can be easily derived from a widespread physical solid property of biological systems and therefore is considered to be a manifestation of this physical property. Indeed, within the QED framework, we can understand that (i) in a coherent system, the components do not respond independently, but as a whole, and (ii) the smaller the stimuli, the more relevant the responses. In particular, when $S<S_{0}$, where *S*_0_ is a stimulus associated with a null response, we obtain *R* < 0, so that the system performs “inward” oriented dynamics that updates the system. A coherent system is adversely affected by strong, powerful inputs simply because they are able to “kick-off” the coherence of their resonating components, whereas a weak or minimal stimulus collectively affects their oscillatory phase attractor [48]. This intriguing property goes hand in hand with the fact that a coherent system cannot be interpreted as a thermal machine. After all, it is well known that the biological thermodynamic yield *η* does not follow the relation $\eta \leq \frac{T_{2}-T_{1}}{T_{2}}$ as postulated by the Carnot cycle—a widely used concept in the field of mechanical engineering). Rather, in biological systems, *η* can reach values up to 80% even though the maximum temperature difference within these systems does not fluctuate more than a few Kelvins. This fundamental thermodynamic aspect that governs living matter is highly relevant in the context of dissipative dynamics. 

Thus, the cyclotronic ion resonance phenomenon testifies to many important facts. First, it shows that the passage of ionic species across cell membranes (where the electric potential is typically 70–100 mV) occurs through a subtle frequency modulation of the oscillating magnetic fields in the cell medium (and we will explain better below how this occurs by reporting on the topological properties of water structures associated with biomolecules in the form of chains, membranes, beads, colloids, etc.). Second, it shows that the passage through the membrane follows a particle-wave duality, in that the wave-function determines the selectivity of the channel. Third, it tells us that the functioning of biological systems is much more sensitive to “subtle” influences (to “minimal stimuli” in fact) than to large, violent stimuli that tend to “stiffen” the sensitive response dynamics of the living state. Fourth, we can understand how biological systems, which are essentially composed of carbon, nitrogen, hydrogen, oxygen, and a number of other elements, may be able to detect and sense very low-intensity fields and use them for their autopoietic dynamics. There are numerous cases that show the sensitivity to constant magnetic fields (DC): for example, about 1 nT in bees and 10 nT in birds [51,52,111]. Now, there is a rational basis for understanding the physical prerequisites recognizing the delicate feature of the electromagnetic balances in biology. The coherent water matrix in living tissue should therefore be regarded as the fundamental transducer as it brings together energy, matter, and information pathways. 

The issues raised above are very intriguing as they elucidate the current misconceptions of living matter (the problem also known as the “K_B_T problem” [112], which has been well disentangled in the framework of QED; for further details, see [113]. In essence it demonstrates the incredibly low energy required along with the extreme efficiency living systems work with. This is also corroborated by the vast number of chemical, electrochemical, electromagnetic, and thermodynamic processes that are performed every second with an extremely low error rate. One of the most energy-consuming of these processes is the ubiquitous ion transport that occurs in almost all biochemical activity. This activity (alone) would cause immense energy dissipation due to the Joule effect that should be associated with the striking ion (and electron) currents that circulate through membranes and the aqueous matrix. Obviously, this is not the case, and the explanations are rooted in electrodynamic coherence. Here, it is noteworthy to mention that ions are involved in energy and charge transfers not just through membranes but also along molecular backbones [42]. If life scientists sincerely wish to understand how water behaves near hydrophilic surfaces and how the dialectic between coherent water, interfaces, and fields can create a topological complexity of 1D, 2D, and 3D systems in both the cytoskeleton as well as in the extracellular matrix, then a real paradigm shift away from the current neo-Darwinistic interpretation is required. 

Finally, water is the main component of living matter (not only in terms of weight but especially in terms of number) and exceeds the amount of all other species by orders of magnitude; in second place are ions (see *E. coli* Table 1 and Table 2 and for humans Table 3). 

Given the numerous arguments presented herein, it is no longer possible to envision biological functioning as if ions behave diffusely according to the Brownian concept. In order to understand the wider picture, we need to analyze how organic molecules configure themselves as a function of water and why coherence is involved to topologically organize living matter. 

## 3. Discussion

The existence of complementary observables highlights an essential uncertainty and implies that at least two quantities are not simultaneously observable. When attempting to explain coherence in macroscopic systems, such as living organisms using QED, the quantum phase operator, $\hat{\Theta }$—conjugated to the number operator, $\hat{N}$ (the smallest entity in a given system at a given scale [28])—emerges as a crucial variable. Given the thermodynamic openness of a coherent system (like a living one), intrinsic fluctuations Δ*N* of its quanta are in force: [71] Δ*N*·Δ*φ* ≥ ½ (in natural units).

The crucial point here is that such an uncertainty relation is scale-invariant as only pure numbers without units are involved. It follows that there is no fundamental difference between observer and observed system between the quantum and classical system. The classical way of thinking results when the total number of quanta can be determined with certainty, i.e., when Δ*N* = 0. In this case, the phase φ is a random variable that varies randomly (incoherent) from one quantum to another. However, if the total number of quanta is not known with certainty, the quantum phase can take a well-defined value (Δ*φ* → 0) if Δ*N* → +∞. This situation describes a coherent quantum regime that is repeatedly observable at the macroscopic level such as in ferromagnetism, ferroelectricity, superconductivity, and superfluidity. This type of quantum coherence is typical for systems with high density and thus also occurs in water in the liquid state. The essential point here is that a “quantum” must not necessarily be an elementary particle, but can be an ion, a molecule, a protein, a cell or even a bird or fish (Figure 9) and at a given scale can be considered as a whole entity, thus cannot be broken down into smaller parts without losing that which led to the essence (behavior). Hence, thinning out via dilution—such as the vapor phase of water, for example—(Δ*N* → 0), implies that coherence will be lost ([71]). It is this kind of quantum coherence in high-density situations that makes it particularly applicable to a living organism [121]. 

The principles outlined here are therefore in perfect agreement with the biphasic reaction pattern as described by hormesis (an excessive energy input leads to decoherence of one or more subcomponents, whereas only a sufficiently small field stimulus promotes coherent coupling between ionic species). The observed hormetic effect could form the basis for many therapeutic applications. Indeed, a sufficiently small, minimal stimulus that can elicit a collective and coherent response of the entire system is nothing less than the most appropriate update that a system’s eigenstate can receive if it wishes to remain consistent with the environmental stimuli it receives [123].

## 4. Conclusions

It is now abundantly clear that living structures can respond to very weak electrical and magnetic signals, some of which in the order of 100 to 1000 times lower than the theoretically predicted lower thresholds. The assumption that the geomagnetic field of our planet enables resonant interaction of ultra-weak electromagnetic fields in living organisms is a given fact when considering existing technical applications (see nuclear magnetic resonance (NMR), electron spin resonance (ESR), and ion cyclotron resonance (ICR)). All of these require the simultaneous application of two fields, one of which is static and the other oscillating. Although the resonance frequencies of NMR and ESR are quite high (due to the coupling to nuclear or electron spin), this coupling in the case of ICR requires much lower frequencies. As demonstrated by Liboff [90,91], the ICR bio-interaction has resulted in relevant medical applications, as ICR makes it possible to manipulate the physiological activity of key biological ionic species such as Ca^2+^, K^+^, Mg^2+^, and other ions upon exposure to calculable magnetic frequencies and intensities. Their interaction with charged particles in living tissues (ranging from basic constituents of the atom to molecules, proteins and polymers to include DNA and RNA) asserts that “life is an expression of the electromagnetic force” [118]. Since ultra-weak magnetic fields are capable of modulating the ratio of the non-coherent vs. coherent fraction of biologically contained water, it is possible to explain many, if not all, of the hormetic effects. This has practical application in both “mild energy” techniques as well as electromagnetic therapies [124].

QED has shown to be the only consistent theory so far to explain these “unreasonable effects” [85] in soft matter. Within this framework, the occurrence of water to be present in two phases provides the basis for the all-or-none responsiveness of living matter upon exposure to extremely low-frequency stimulation. Indeed, by considering each single degree of freedom (for instance, the electronic oscillation of water molecule between sp^3^ and 5d molecular orbitals, responsible for the emergence of the condensed liquid phase and for the amorphous phase in the solid), the ratio between the coherent (F_C_) vs. incoherent (F_nC_) fraction must neither “freeze” the homeostatic dynamic within biota into immobility (F_C_ = 1) nor deteriorate into entropy (F_C_ = 0). Both extremes would result into the cessation of the living state. As depicted in Figure 3, viability for most living species is assured only when the labile equilibrium between these extreme poles is maintained within a quite narrow (viable) temperature band (comparable to the habitable zone determined by the distance of Earth to the Sun). However, in biological matter, a huge array of coherences (on several degrees of freedom) are in force at the same time in a nested hierarchy, which implies that no process in living matter is assigned to “chance”. The momentaneous exit out of one kind of coherence for a certain molecule, is, however, accompanied by its contemporaneous belonging to many other levels of coherence. For instance, when an enzyme is rhythmically changing its own steric configuration on a time-scale of the order of 10^−6^ s, it is also participating with something like 10^13^ reactions per second, and all these processes are possible because, for instance, some of its hydration water molecules are made to pass from a coherent to a non-coherent state on certain degrees of freedom (like the dipole oscillation or their rotation or the other coherences played by possible ions). This to say that living matter is a “device which generates order from order”, responding non-randomly to the, possibly random, stimuli coming from the world. This implies a non-random adaptation, written in the thermodynamic history of coherence and coupling with the environment.

Moreover, the electrodynamic approach maintains a natural electromagnetic interaction between the field and the particles involved (whole set of oscillatory frequencies for nucleic acids, transcription, RNAs, enzymes, etc.), so as to orchestrate biochemical reactions (avoiding biochemical pathways that are not required), thus tuning coherent groups of cells to a complex hierarchy of different frequencies to serve and maintain the life process of the organism. Needless to say, this kind of electromagnetic “whispering” is predominantly a physically driven process that manifests itself chemically as a result of the autopoietic turnover of matter. Thus, QED has the potential to explain not only the cellular dynamics of biochemical pathways, but also it is so far the only tool that can assist in fully explaining the phylo-ontogenetic dynamics during morphogenesis (from the zygote to the fully formed juvenile stages) as well as tissue regeneration during injuries (stretching from wound healing to the regrowth of lost extremities) that rely on extremely weak electrical stimulation [125,126,127,128]. Moreover, the effects documented by Ho et al. [129] demonstrating that weak magnetic field exposure (<1 μT) in the SLF-band induce abnormal larval development in *Drosophila melanogaster*, becoming comprehensible when interpreted within the QED framework. Furthermore, its relevance in higher organisms (e.g., humans) is even more pronounced as these are considered “super-organisms” (symbiosis of non-human and human cells) forming a unique ecosystem that maintain complex biologic functions that also rely on electromagnetic homeostasis [130]. This implies that, upon exposure during sensitive windows of development with appropriate technical field frequencies, it increases the likelihood of adverse effects as these self-maintained attractors experience a steady drift away from the healthy state towards a new one that manifests metabolic dysfunction and lower resilience. In essence, these issues shed light onto the basic mode of action by which biota are affected by adverse electromagnetic exposure, in that technical fields (including low-frequency modulation, such as those used in RF bands of Wi-Fi and mobile telephony) have a direct impact on the biogenic morphogenetic field. Thus, the disturbance of regeneration or growth of tissues by incompatible EM fields affects foremost the “biogenic” field of the organism, which in turn reverberates as an epiphenomenon onto the biomolecular dynamics, altering biomolecular processes. If this disturbance is not disruptive to coherence, it has only a temporaneous effect, as the system, being still coherent, is robust enough to develop an adequate physiology and to maintain homeostasis even in the presence of this (non-biological) electromagnetic distressing stimulus. Thus, the QED perspective would explain the multitude of discordant and contradictory studies relating to the adverse health effects of EMF exposure [131]. Of course, this comes at a price: costly energetic adaptation, which is important to consider as it reflects the persisting “intelligent” response pattern for as long as the system is alive (i.e., coherent), within which no random processes are possible. Since coherence is thermodynamically convenient (deepening the energy gap), it is strictly tied to any open system. Another important aspect concerns the impossibility of reproducing emergent properties of a living organism by simply isolating and analyzing parts of it. The existence of a hierarchy of coherence implies that molecules, electrons, ions, as well as collective groups of embedded substances, behave in a completely different way as soon as they are involved in multiplexed in-phase correlations than when this is not the case. This also means that results from experimental studies performed in vitro (including research on the effects of electromagnetic radiation on cells and tissues) cannot be extrapolated to in vivo systems *tout court*. Indeed, the supercoherent organization forces the in-phase oscillators to first overcome an energy gap before they can behave independently, i.e., become decoherent.

## Figures and Tables

**Figure 1 ijms-24-14003-f001:**
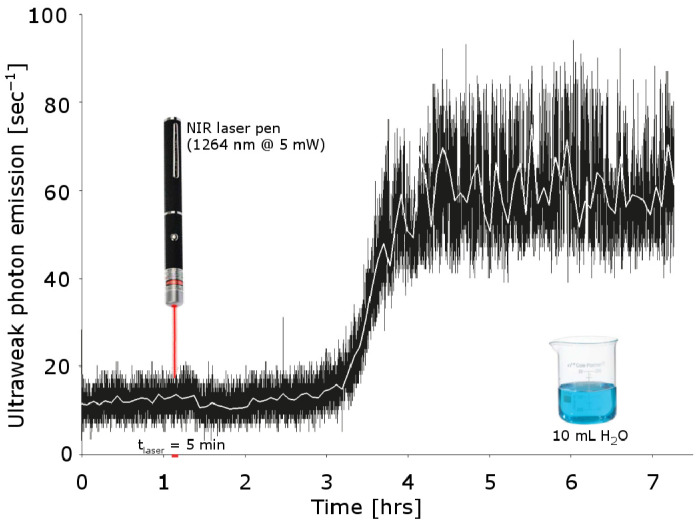
Laser-induced chain reaction in water. A 5 min exposure of air-saturated bidistilled water to low-intensity laser (5 mW) infrared radiation (with 1264 nm) induces, after a long latent period, luminescence in the blue-green region that last many hours. The pumping activity is triggered as the laser wavelength corresponding to the wavelength of the electronic transition of dissolved oxygen inducing a transition from the triplet state into the singlet state (compiled and adapted from [47]).

**Figure 2 ijms-24-14003-f002:**
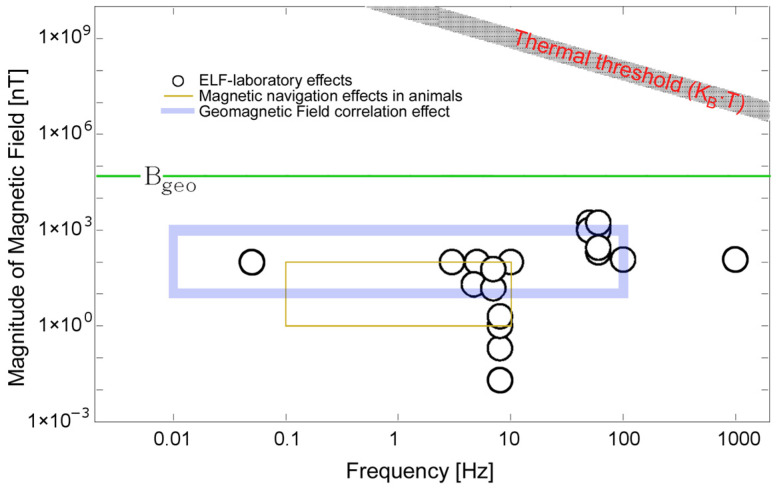
Hormetic effects of extremely low-frequency stimulation with ultra-weak magnetic field intensities plotted as a frequency–amplitude diagram. Parameters of super-weak magnetic fields supposedly causing observed biological effects, yet still being well below the thermal threshold level. The amplitude–frequency intervals are approximate orders of magnitude (redrawn and adapted from [51,52]).

**Figure 3 ijms-24-14003-f003:**
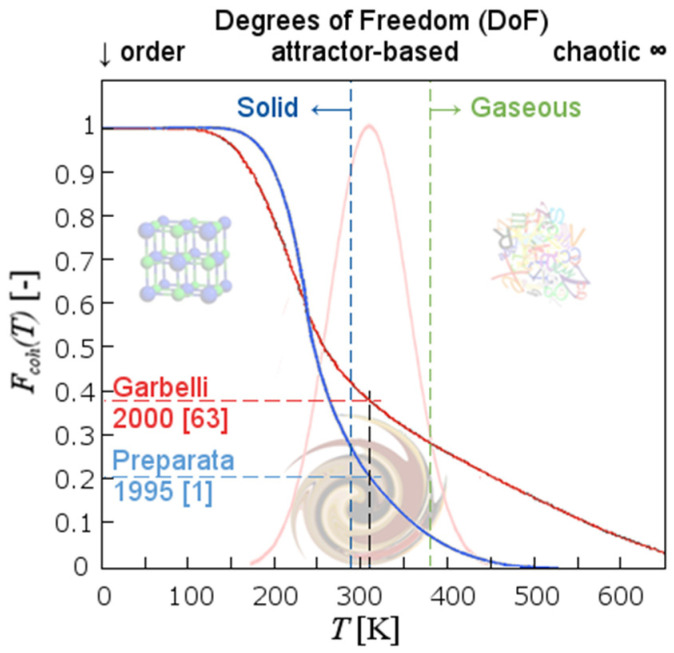
Calculated trend of the “coherent fraction” F_Coh_(T) vs. temperature. Water at sub-zero temperatures becomes almost fully coherent, F_Coh_(T) ≈ 1, thus ordered (few degrees of freedom (DoFs), whereas water in the vapor phase asymptotically approaches F_Coh_(T) = 0 (DoFs approach an infinitely large number). The original calculations showed that at physiological temperatures (310 K) the coherent fraction is 20% (dashed blue line) [1]. A revision and finetuning of the relevant parameters involved revealed that the coherent fraction in warm-blooded animals amounts to even 40% (dashed red line) [63]. In this context, it was also found that coherent elements can still be found even at temperatures > 600 K. The optimal viable window, however, where most organisms thrive, as highlighted by the cub-imposed Gaussian distribution, is found in-between the solid and the gaseous states (with limited DoFs, embraced by the dashed blue and dashed green line).

**Figure 4 ijms-24-14003-f004:**
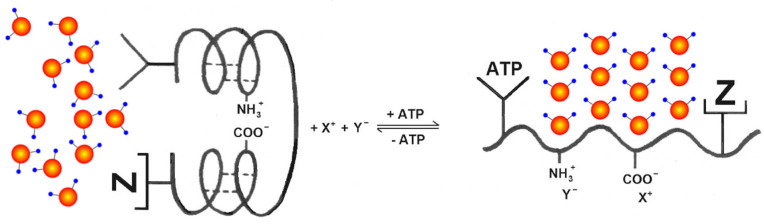
Schematic illustration of how binding and adsorption of ATP along with its “helpers” (congruous anions and protein X, shown as Z) unravels the “folded” secondary structure (**left**). Selective K^+^ adsorption on the liberated β^−^, and γ^−^ carboxyl groups occurs, triggering multilayer water polarization on the exposed backbone NHCO groups characterizing the resting state of the protein (**right**) (modified after [66,68]).

**Figure 5 ijms-24-14003-f005:**
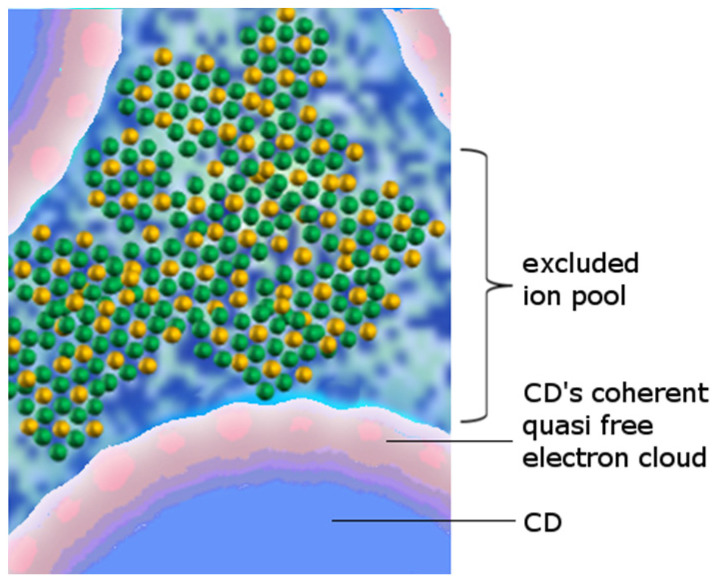
Sketch showing the state of dissolved ions in liquid water according to QED theory. The coherent, non-ideal Debye–Hückel plasmas located in the non-coherent fraction of water explain how such a stable system—like the ionic crystal—can dissolve and why with the increase in temperature increases the solubility of many ionic species.

**Figure 6 ijms-24-14003-f006:**
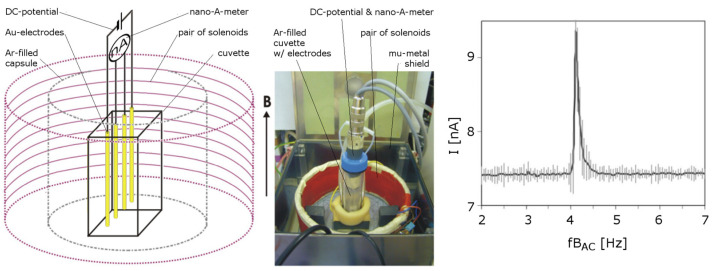
An example of the experimental setup for the realization of the Zhadin effect. Left: cell scheme; center: photo of the real setup; right: current peak in glutamic acid solution (in 2.24 mM HCl, pH = 2.85); resonance peak occurred at 4.2 Hz obtained with a static (*B_dc_* = 40 μT) and alternate magnetic field amplitude (*B_ac_* = 50 nT). Applied electric potential between electrodes: 80 mV. Figure adapted and modified after Pazur [88].

**Figure 7 ijms-24-14003-f007:**
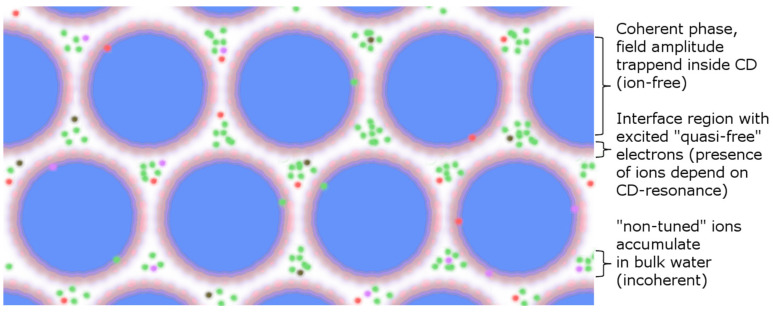
Schematic of the water CDs together with the intermediate CD space (“reservoir” zone) where the dissolved ions gather (grouped colored dots). The resonance attraction between CDs and ionic species acts like a selective ion-filter and is predominant for ions located in the “interface” region (single dots).

**Figure 8 ijms-24-14003-f008:**
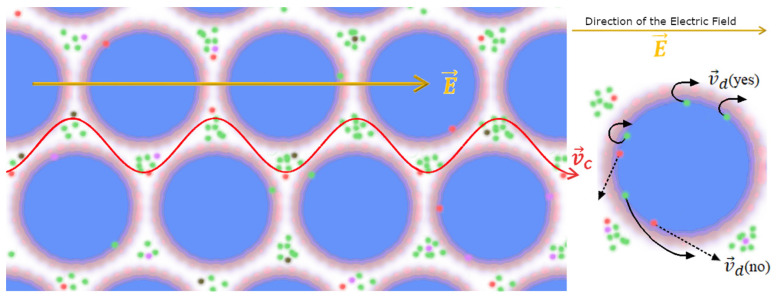
Interfacial region of a CD and possible ion trajectories under the influence of fields. **Left**: Only in the presence of weak fields can the drift velocity be appropriate so as to allow a deflection of the ion trajectory within the interfacial zone, ${\vec{v}}_{d}\left(yes\right)$; otherwise, the ion falls back into the reservoir zone and no longer contributes to the current, ${\vec{v}}_{d}\left(no\right)$. The same holds for excessive field intensities that cause the ions to fall into the bulk reservoir of the non-coherent water fraction. **Right**: top view of suitable field intensities resulting in an oscillatory ion drift path of ionic species (colored dots) within the interface region causing ${\vec{v}}_{d}\left(yes\right)\to {\vec{v}}_{c}$ to ultimately contribute to the ion current peak; modified and adapted after [85].

**Figure 9 ijms-24-14003-f009:**
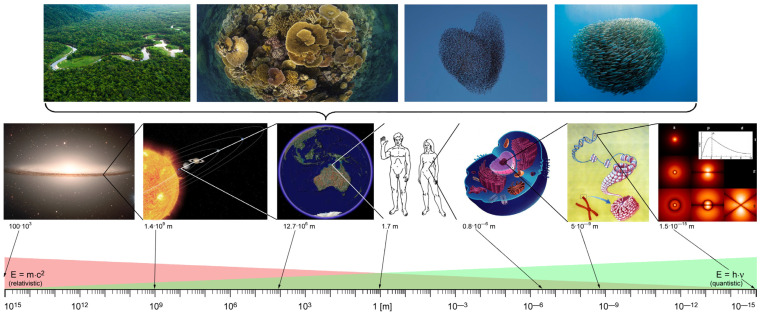
Collective many-body coherent behavior as an emergent property seen on each of the presented hierarchical levels. The “fundamental uncertainty relationship“ (Δ*N·*Δ*φ* ≥ ½) becomes especially manifest in the biologic realm and is most visible in swarm of birds and fishes as well as coherent coupling in larger biomes, e.g., reefs and forests of the tropics. QFT states that the total number of quanta N and the common phase angle φ of a swarm is an emergent consequence of that relationship (adapted from [67,122]).

**Table 1 ijms-24-14003-t001:** The chemical composition of a common bacterium such as *E. coli* with a dry mass of about 0.95 pg [114]. The weight and molar fraction of chemical species are given, and a final count of their number in the last column shows that water and ions outnumber by orders of magnitude any other cytosolic component [115].

Matter	Molar Weight (Da)	Mass Fraction (%)	Moles (mM)	Number (−)	Mol Fr. (%)
Water	18	70	3888.9	23,400,000,000	99.324
Ions	47	1	300	120,000,000	0.509
Amino Acids	110 (av.)	2	1.5	6,000,000	0.025
Nucleotides	414 (av.)	0.8	1.9	3,530,000	0.015
Proteins	40,000 (av.)	15	0.34	3,600,000	0.015
RNA	33,500	6	0.18	222,000	0.001
Polysaccharides	1,000,000 (av.)	3	0.003	39,000	0.001
DNA	2,840,000,000	1	0.000005	1	0.000
Others	-	0.4	-		
Total	3,159,782	100		23,533,391,001	100.00

**Table 2 ijms-24-14003-t002:** Details of the ion composition of *E. coli*, molar concentration, and total number (sorted by abundance). Potassium alone is far more abundant than the other ions combined, which underlines that ion transport is one of the most important processes in metabolic functions of any organism at any phylogenetic level [115]. A detailed list can be found in Ch.3, p.131ff in [116]. Cyclotron resonances are reported from [117,118] (those flanked by an asterisk represent theoretical values only; furthermore, no values are assigned to anions as only cations are able to co-resonate in mixed water CDs. The case of NH_4_^+^ is special as it is usually present in the extracellular region and only assimilated during certain metabolic processes).

Ion	Concentration (mM)	Number (−)	Number (%)	f_ICR_/B (Hz/μT)
Potassium (K^+^)	200–250	90,000,000	69.147	0.194
Iron (Fe^2+^/Fe^3+^)	18	7,000,000	5.378	* 0.546
Bicarbonate (HCO_3_^−^)	12	5,670,000	4.356	-
Chloride (Cl^−^)	6	5,050,000	3.880	-
Magnesium (Mg^2+^)	10	4,000,000	3.073	1.255
Hydronium (H_3_O^+^)	10–4.2	3,000,000	2.305	* 0.802
Zündel cation (H_3_O^+^∙H_2_O)				* 0.477
Trimer hydronium (H_3_O^+^∙2H_2_O)				* 0.277
Tetramer hydronium (H_3_O^+^∙3H_2_O)				* 0.209
Magic hydronium cation (H_3_O^+^∙20H_2_O)				* 0.167
Calcium (Ca^2+^)	6	2,300,000	1.767	0.761
Dihydrogen phosphate (H_2_PO_4_^−^)	5	2,107,700	1.619	-
Sodium (Na^+^)	5	2,000,000	1.573	* 0.663
Cuprum (Cu^2+^)	4	1,700,000	1.306	* 0.480
Manganese (Mn^2+^)	4	1,700,000	1.306	* 0.555
Molybdenium (Mo^4+^)	4	1,700,000	1.306	* 0.636
Zincum (Zn^2+^)	4	1,700,000	1.306	* 0.466
Phosphoenol-Pyruvate (PEP^3−^)	2.8	1,100,000	0.845	-
Pyruvate (CH_3_COCOO^−^)	0.9	380,000	0.292	-
Adenosin-Diphosphate (ADP^3−^)	0.63	270,000	0.207	-
Nicotinamide (NADP^3−^)	0.63	240,000	0.184	-
–”– (NADPH^4−^)	0.56	220,060	0.169	-
Ammonium (NH_4_^+^)	-	-	-	* 0.845
Glucose-6-Phosphate (6GP^2−^)	0.05	20,000	0.015	-
Proton (H^+^)	0.000063	30	0.000	15.13
Total		125,077,820	100.00	

(*) personal communication [118].

**Table 3 ijms-24-14003-t003:** Average body composition for an adult Caucasian female (♀) and male (♂) within the age slot 20–29 years of the United States [119,120].

	m (g)		Molar Weight (Da)	Weight (%)		Mol (%)		Mol (%)	
	♀	♂		♀	♂	♀	♂	♀	♂
Water (∑)	30,900.0	45,000.0	18.0	52.32	55.14	96.07	96.49	-	-
–”– intracellular	15,400.0	24,900.0	18.0	26.07	30.51	47.88	53.39	96.71	97.39
–”– extracellular	15,500.0	20,100.0	18.0	26.24	24.63	48.19	43.10	-	-
Lipids	15,900.0	19,800.0	600.0	26.92	24.26	1.48	1.27	3.00	2.32
Proteins	9720.0	13,070.0	40,000.0	15.70	16.01	0.01	0.01	0.03	0.02
Phosphates	1372.0	1704.0	98.0	2.32	2.09	0.78	0.67	-	-
Calcium	905.5	1230.0	40.1	1.53	1.51	1.26	1.18	-	-
Carbohydrates	500.0	500.0	1,000,000.0	0.85	0.61	0.00	0.00	-	-
Potassium	88.5	149.0	39.1	0.15	0.18	0.13	0.15	0.26	0.27
Sodium	63.5	82.0	23.0	0.11	0.10	0.15	0.14	-	-
Chlor	61.5	77.5	35.5	0.10	0.09	0.01	0.08	-	-
Total	59,511.0	81,612.5		99.99	99.99	99.93	100.00	100.00	100.00

## Data Availability

Not applicable.

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
