# Peer review of "Quantum Electrodynamics Coherence and Hormesis: Foundations of Quantum Biology"

_ijms, 2023, doi:10.3390/ijms241814003_

Round 1
Reviewer 1 Report
Referee report on the Review Article, "QED Coherence and Hormesis: foundations of Quantum Biology"
by Pierre Madl and Paolo Renati:
--------------------------------------------
The most important achievements in several fields of science, have been reached when one could explain the observed phenomena and properties of Nature, by a deeper fundamental theory for the description of them and with new predictions. To explain thermodynamics by statistical theory, electrostatics and magnetism by the Maxwell theory, atomic physics and chemistry by Quantum Mechanics and eventually by Quantum Electrodynamics (QED) , superconductivity and superfluidity by Quantum Field Theory (QFT)/QED, to mention a few of them. Among them, a very important field has been Quantum Optics and a seminal one in it, Laser Physics, which has found innumerable applications, e.g. in medicine.
The submitted Review Article is among one of the attempts to present several ideas along the same lines to eventually reach a microscopic/fundamental theory of Quantum Biology. The attempts in this direction are among the most important and useful ones in Life Sciences, one of which is Quantum Biology.
The presentation in the submitted Review Article is transparent and everything is presented patiently and quite clearly, to the extent that even a "layman" can get interested in the subject. At the same time the numerous references given in the paper are very well chosen and indeed are necessary for any review article to be self-contained.
However, as usual this article can still be improved, since a few very important references on one the main subjects of the paper, namely, on "coherence" in my opinion are missing. Adding a few new references, which have shown (see below) the absolute necessity of coherence for some phenomena to happen, would emphasize the importance as the ingredients not only in Quantum Biology, but also in some other fundamental theories and phenomena. That would also make the Review Article more comprehensive and would encourage more the theoretical and mathematical physicists to get interested in Quantum Biology.
My recommendation to the Authors is to add in the section on Coherence the following three articles:
R.J. Glauber, Coherent and incoherent states of the radiation field,Phys. Rev. 131 (1963) 2766.
J. R. Klauder and G. Sudarshan. Fundamentals of Quantum Optics. Dover Books in Physics (2006).
A.Tureanu, Neutrino Oscillations by a Manifestly Coherent Mechanism and Massless vs. Massive Neutrinos, arXiv:2304.13491 (hep-ph); https://www.sciencedirect.com/science /article/pii/S0370269323003301, Phys. Lett. B. (2023) in print. The work by Glauber is the one on the theory of Laser Physics, bringing Nobel Prize to him. The other paper by Klauder and Sudarshan is a book which contains a full and mathematically rigorous description of coherence, coherent states as ones of the fundamentals in Quantum Optics. The work by Tureanu is the latest work and the most exact (and actually, the correct one within QFT) description of Neutrino Oscillations, which brought the experimentalists to Nobel Prize in 2015 for its observation, and besides the QFT techniques used in it is as the ones Bogoliubov used to describe the BCS (Bardeen-Cooper-Schrieffer) first theory of superconductivity, in a most transparent and rigorous way. Besides, it contains a list of the previous important references on the subject. After the inclusion of the above references, I shall be glad to recommend the Review Article by Pierre Madl and Paolo Renati for publication in the Special Issue of IJMS.
Author Response
Dear Reviewer,
Thank you very much for your feedback regarding our manuscript entitled “QED Coherence and Hormesis: the foundations of Quantum Biology” intended for the special issue within the “International Journal of Molecular Sciences” entitled "The Emerging Role of Quantum Sciences and Radiation Biology in Biomedical Applications".
In order to keep it short we directly address the issues raised by you:
Replies to revisor 1:
As suggested to make our review article more comprehensive and updated we have included additional reference about coherence. We accepted to include the advised references, in order to attract attention of more theoretical and mathematical physicists to become engaged in Quantum Biology.
We included a reference of Roy Glauber who addressed coherence in relation to laser physics
Glauber, R. J. (1963). Coherent and Incoherent States of the Radiation Field. Physical Review, 131(6), 2766–2788. doi:10.1103/physrev.131.2766
Furthermore, we also cited the classic in Quantum Optics – however in doing so we could only access the 1968-edition. Nonetheless, the rigorous description of "coherent states", discussed in this monograph makes this a valuable contribution
Klauder JR, Sudarshan G (1968) Fundamentals of Quantum Optics. WA Benjamin, Inc., New York (USA)
On top of that, we completed the list of added references with a contribution regarding the Bogoliubov transformation
Bogoliubov NN, Shirkov DV (1983) Quantum Fields. Benjamin/Cummings Publ. Inc. Massachusetts (USA) ISBN 0-8053-0983-7
Being aware that these references will go well-beyond the comprehensibility of most biologist, we hope these may indeed become an incentive for physicists specialized in QFT to address quantum phenomena in biology in as much as prof. Herbert Fröhlich initiated back then over four decades ago.[1]
Anyhow, and in accordance with reviewer 1 we decided to upgrade figure 9 to accommodate also the subatomic domain, yet at the same time, based on the principles of the fundamental uncertainty relationship, we embraced also the «galactic dimension».
Finally, as proposed by Rev.1, we added the following three references – these shall be also considered as a continuation to comment “b” of Rev.2 :
Ho MW (1997) Quantum coherence and conscious experience. Kybernetes, 26(3), 265–276. https://doi.org/10.1108/03684929710163164
Ho MW (2017) Meaning of Life and the Universe. World Scientific. Singapore. ISBN 978-981-3108-86-8. https://doi.org/10.1142/10012
Ho MW (2016) Quantum Jazz. I-SIS Lecture. Science in Society: https://www.i-sis.org.uk/QuantumJazz.php (accessed on 05 June 2023)
Regarding the suggested reference made by Rev.1. that deals with Neutrino oscillations:
Tureanua A (2023) Neutrino oscillations by a manifestly coherent mechanism and massless vs. massive neutrinos. Physics Letters B 843: 137996. doi: 10.1016/0375-9474(69)90295-4
We would like to acknowledge that in our view this paper far exceeds the contextual relevance that we are dealing with. In addition, with mechanisms at subatomic scales, there is no need to involve large numbers of entities as is the case with the aqueous phase in biota or in condensed matter, respectively. We want to highlight that the negative coupling term between EM-field and oscillating charge (which in water results in the energy gap once it slips into the coherent state) when considered for single molecules is rather insignificant (i.e., 1 ppm with respect to the energy at which the electron is placed on its orbital) as this tiny negative energy will not be sufficient to give rise to a new stable state (i.e.: a phase transition). Only once a critical density is reached with a sufficiently high number – like those involved in a CD (i.e., several million water molecules) – this negative energy becomes large enough to drive the system into a new – lower, thus stable – ground state where the phase of oscillation becomes well defined and common to all the oscillators coupled to the field. As biota are open systems, all the participating molecules expel the excess energy back to the vacuum, as to become part of the coherent ensemble.
[1] Fröhlich H (1978) Coherent Electric Vibrations in Biological Systems and the Cancer Problem. IEEE Transactions on Microwave Theory and Techniques, 26(8): 613-618; https://doi.org/10.1109/TMTT.1978.1129446
Fröhlich H, Kremer F (1983). Coherent Excitations in Biological Systems, Springer, Berlin (FRG). ISBN 978-3-642-69188-1. https://doi.org/10.1007/978-3-642-69186-7
Fröhlich H (1986) Coherent Excitation in Active Biological Systems. In: Gutmann F, Keyzer H (eds) Modern Bioelectrochemistry. Springer, Boston, MA. https://doi.org/10.1007/978-1-4613-2105-7_8
Hoping that all issues have been addressed properly, we thank you for your constructive feedback.
Sincerely,
Dr. Pierre Madl
Dr. Paolo Renati

Reviewer 2 Report
This paper studies the possibility of applying the mathematical concepts and techniques of Quantum Field Theory (QFT) to biology. The motivation for this treatment is drawn from authors’ belief that the dynamical order found in living systems can be properly understood only through a theoretical framework that allows to accommodate such key concepts like coherence, dissipation or symmetry breaking (phase transitions) and ensuing distinct vacua (phases).
The topic of the paper is interesting, and it may be that the authors have results to intrigue a wide swath of the IJMS readership. However, the paper as written is densely worded and lacks mathematical rigor. This is in essence a mathematical question and it might be a matter of taste, but for people from a physics community (who should also be the recipients of the paper’s message) this can be a bit annoying. I should also stress that from the physics point of view, the language used is a bit vague and almost archaic.
I do not feel much competent to address the biology part of the paper, but there are few issues with the physics that need to be sorted out before it can be published. These are:
a) Sentence on lines 128-130: “…the condensation of NG bosons relative to the degrees of freedom at which the symmetry is broken underlies the appearance of order parameters.” doesn’t make much sense. What type of the symmetry breakdown the authors consider? The Nambu-Goldstone theorem about existence of massless NG particles is associated with spontaneous symmetry breakdown (i.e. 2nd order phase transition). Similarly, discrete symmetry breakdown, such as Z_2 transition does not have any NG associated with the transition. The authors should be clear (throughout the whole paper) what type of the symmetry breakdown they consider in the paper.
b) Since the aim of the paper is to discuss symmetry breakdown in nonrelativistic systems (living systems), it should be stressed that the simplistic NG counting (i.e., how many NG fields are related to number of broken symmetries) employed by the authors involves a number of subtleties. In Lorentz invariant theories it turns out to be exactly one NG boson for each broken symmetry generator. Eg., in condense matter systems, however, it is known that there might be less on (or none) NG fields for a give number of broken symmetries. Typical example is provided by antiferromagnets (2 NG) vs. ferromagnets (1 NG) where in both cases the SU(2) symmetry of spin rotations is spontaneously broken by mutual spin alignment to the U(1) subgroup (2 broken symmetry generators).
c) Line 161, the statement “… order is lack of symmetry …” is very misleading. Transitions in thermodynamic systems are from ordered to disorder phases (as temperature lowers), where the ordered phase is precisely the phase with higher symmetry group than the disordered phase.
d) Discussion, lines 734-736, “If all physical quantities would be simultaneously measurable, one would soon realize the existence of a “state variable” … However, if there are quantities that are not simultaneously measurable, then a state variable is illusionary.” This statement has strictly no sense in quantum theory. State is defined in quantum theory (both in QM and QFT) by “state variables” which correspond to a “complete set of observables”. Complete set of observables is a maximal set of pairwise commuting observables (i.e., corresponding operators). Quantum state described by a complete set of observables provides a maximal available information on the system. In this respect state variable is not illusory – quantum state is defined by the values of its state variables.
e) Line 738-739, sentence “The absence of state variable highlights an essential uncertainty and implies that at least two quantities are not simultaneously observable.” should read “The existence of complementary observables highlights …”.
All in all, the presented manuscript would deserve publication as a paper in IJMS, provided the authors incorporate the above comments to the revised manuscript.
Author Response
Dear Reviewers,
Thank you very much for your feedback regarding our manuscript entitled “QED Coherence and Hormesis: the foundations of Quantum Biology” intended for the special issue within the “International Journal of Molecular Sciences” entitled "The Emerging Role of Quantum Sciences and Radiation Biology in Biomedical Applications".
In order to keep it short we directly address the issues raised by you:
Replies to Revisor 2:
Given the special points addressed by the reviewer’s we would like to respond to each of the four questions in the following way:
R.2 - a) Sentence on lines 128-130: “…the condensation of NG bosons relative to the degrees of freedom at which the symmetry is broken underlies the appearance of order parameters.” doesn’t make much sense. What type of the symmetry breakdown the authors consider? The Nambu-Goldstone theorem about existence of massless NG particles is associated with spontaneous symmetry breakdown (i.e. 2nd order phase transition). Similarly, discrete symmetry breakdown, such as Z_2 transition does not have any NG associated with the transition. The authors should be clear (throughout the whole paper) what type of the symmetry breakdown they consider in the paper.
Authors: Thank you for highlighting this issue. We reformulated the statement (lines 128-130). However, we try to explain what is the essence of our concepts, by referring to an extract from reference 4 of our manuscript (accessible at: arXiv:quant-ph/9609014v1 18 Sep 1996) where prof. Giuseppe Vitiello writes:
[…. «As well known, when some kind of atoms (or molecules) sit in some lattice sites we have a crystal. The lattice is a specific geometric arrangement with a characteristic length (I am thinking of a very simple situation which is enough for what I want to say). A crystal may be broken in many ways, say by melting it at high temperature. Once the crystal is broken, one is left with the constituent atoms. So the atoms may be in the crystal phase or, e.g. after melting, in the gaseous phase. We can think of these phases as the functions of our structure (the atoms): the crystal function, the gaseous function. In the crystal phase one may experimentally study the scattering of, say, neutrons on phonons. Phonons are the quanta of the elastic waves propagating in the crystal. They are true particles living in the crystal. We observe them indeed in the scattering with neutrons. As matter of fact, for the complementarity principle, they are the same thing as the elastic waves: they propagate over the whole system as the elastic waves do (for this reason they are also called collective modes). The phonons (or the elastic waves) are in fact the messengers exchanged by the atoms and are responsible for holding the atoms in their lattice sites. Therefore, the list of the crystal components includes not only the atoms but also the phonons. Including only the atoms our list is not complete! However, when you destroy the crystal you do not find the phonons! They disappear! On the other hand, if you want to reconstruct your crystal after you have broken it, the atoms you were left with are not enough: you must supplement the information which tells them to sit in the special lattice you want (cubic or else, etc.). You need, in short, to supplement the ordering information which was lost when the crystal was destroyed. Exactly such an ordering information is “dynamically” realized in the phonon particles. Thus, the phonon particle only exists (but really exists!) as long as the crystal exists, and vice versa. The function of being crystal is identified with the particle structure! As you see there is a lot in the quantum theory of matter and please notice: the description of crystal in terms of phonons has nothing to do with “interpretative problems”. It is a well understood, experimentally well tested physical description.
Such a situation happens many times in physics; other familiar examples are ferromagnets, superconductors, etc.. It is a general feature occurring when the symmetry of the dynamics is not the symmetry of the states of the system (symmetry is spontaneously broken, technically speaking).
Let me explain what this means. Consider the crystal as an example: the symmetry of the dynamics is the continuous space translational symmetry (the atoms may move around occupying any position in the available space). In the crystal state however such a symmetry is lost (broken) since the atoms must get ordered in the lattice sites; they cannot sit, e.g., in between two lattice corners: order is lack of symmetry! A general theorem states that when a continuous symmetry is spontaneously broken, or equivalently, as we have just seen, an ordered pattern is generated, a massless particle is dynamically created; this particle (called the Nambu-Goldstone boson) is the phonon in the crystal case. Please, notice that this particle is massless, which means that it can span the whole system volume without inertia, which in turn guaranties that the ordering information is carried around without losses and that the ordered pattern is a stable one since the presence (or, as we say, the condensation) of the Goldstone particles of lowest momentum does not add energy to the state (it is enough to consider the lowest energy state, namely the ground state); in conclusion, the ordered ground state has the same energy of the symmetric (unordered) one (we call it normal ground state): they are degenerate states. This is why the crystal does exist as a stable phase of the matter. Actually, ground states, and therefore the phases the system may assume, are classified by their ordering degree (the order parameter) which depends on the condensate density of Goldstone quanta. We thus see that by tuning the condensate density (e.g. by changing the temperature) the system may be driven through the phases it can assume. Since the system phases are macroscopically characterized (the order parameter is in fact a macroscopic observable), we see that a bridge between the microscopic quantum scale and the macroscopic scale is established. All the above is of course possible only if the mathematical formalism provides us with many degenerate, but physically inequivalent, ground states which we need to represent the system phases, which in fact have different physical properties: this is why we have to use QFT and not QM, as I said above. In QM, all the possible ground states are physically equivalent (the Von Neuman Theorem); QFT is on the contrary much richer, it is equipped with infinitely many, physically inequivalent ground states and therefore we must use QFT to study systems with many phases. Above I have been mentioning “theorems”: however, I want to stress that these mathematical theorems perfectly fit and are fitted by real experiments and they represent the only available quantum theory (QFT indeed) on which the reliable working of any sort of technological gadget around us is based; in spite of the many epistemological and philosophical unsolved questions quantum theories may arise.
Now you see why I said that I need to start by considering actual material: this is not simply a list of constituents, it is not simply specific information from punctual observations, it is not simply a lot of real data and statistics, but it is also the dynamics. […] There is no hope to build up a crystal without the long-range correlations mediated by the phonons: if you try to fix up atom by atom in their lattice sites holding them by hooks you will never get the coherent orchestra of vibrating atoms playing the crystal function. This is what experiments tell us.» ….]
On top of that and to list examples of various kinds of symmetry breakdown we included to the already listed references (references 10 & 19) additional two references:
Del Giudice E, Doglia S, Milani M, Vitiello G (1985) A quantum field theoretical approach to the collective behavior of biological systems. Nucl. Phys. B, 251(13), 375-400. https://doi.org/10.1016/0550-3213(85)90267-6
Freeman W, Vitiello G, (2008) Dissipation, spontaneous breakdown of symmetry and brain dynamics. J. Phys. A: Math. Theoret., 41(30), 304042. https://doi.org/10.1088/1751-8113/41/30/304042
R.2 - b) Since the aim of the paper is to discuss symmetry breakdown in nonrelativistic systems (living systems), it should be stressed that the simplistic NG counting (i.e., how many NG fields are related to number of broken symmetries) employed by the authors involves a number of subtleties. In Lorentz invariant theories it turns out to be exactly one NG boson for each broken symmetry generator; e.g., in condense matter systems, however, it is known that there might be less on (or none) NG fields for a give number of broken symmetries. Typical example is provided by antiferromagnets (2 NG) vs. ferromagnets (1 NG) where in both cases the SU(2) symmetry of spin rotations is spontaneously broken by mutual spin alignment to the U(1) subgroup (2 broken symmetry generators).
Authors: In biological systems, a lot of degrees of freedom over the components are engaged in coherent dynamics and many kinds of symmetry are thus broken (not even catalogued, known… among which we can guess: oscillations of many molecular species, ions and electrolytes, membranes, enzymes and proteins folding and unfolding, microtubules pulsations etc, etc. and… however a whole phase correlation over the ubiquitous water matrix is in force, like in a jazz ensemble, as, indeed, referred Mae-Wan Ho[1] by comparing biological coherence to a “quantum jazz”. These types of order imply “breakings of the symmetry” in the sense that some kind of invariance (like translations in space and time) is no more valid because, like in crystal lattices, to move along one direction or along another is not equivalent…it’s not symmetric.
We cannot enter too deep into NG condensations (analysing the many possible cases) in this paper. Yet, the main idea that we want to transmit is a “vision reframe” which implies to understand that:
Biological matter is established on coherence.
- Coherence implies special kind of correlation (where the operator “phase” becomes fully defined, at detriment of the conjugated operator “number”) and order.
- Order emerges as “lack of symmetry”, especially in the kind of “order without repetition” (as proposed by prof. Erwin Schrödinger) typical of living structures, in the sense that a system fully symmetric is a fully chaotic one, where here or there, now or then, does not make any difference, while oppositely a system where many symmetries have been broken is more and more ordered (not only in space, but also – and above all – in time: motional order).
This said, please, suggest us, whether it’s better not to mention NG bosons as mediators of symmetry breakings in condensed matter because not generally valid, or else. Thank you.
R.2 – c) Line 161, the statement “… order is lack of symmetry …” is very misleading. Transitions in thermodynamic systems are from ordered to disorder phases (as temperature lowers), where the ordered phase is precisely the phase with higher symmetry group than the disordered phase.
Authors: we reported this statement from what prof. G. Vitiello wrote (see text quoted in our reply “a” to Rev.2). This simplification stems from an interpretation whereby the dynamics which regulates the behaviour of the elementary components of a physical system as to generate the formation of ordered structures, can be found in the physics of elementary particles, of condensed matter, as well as in cosmology and in biological systems. Thus, we simplified by stating that order is lack of symmetry. Given that this generalization in thermodynamic systems – as is the case with biota – requires accentuation in as much as to follow an interpretation in Pitch’s paper “The vacuum expectation value va is called an order parameter of the symmetry breaking …. Notice that we have proved the existence of massless Nambu–Goldstone modes without making use of any perturbative expansion. Thus, the theorem applies to any (Poincare-invariant) physical system where a continuous symmetry of the Lagrangian is broken by the vacuum, either spontaneously or dynamically.”[2] In accordance, we modified our statement by writing that “lack of symmetry due to continuous symmetry breaking is the result of the guided pathways in biotic systems that seeks to maintain a viable homeostasis.” From a biologist’s perspective it can be phrased as follows: …. when the donkey is hungry, and this happens sooner or later, no one will be able to stop him from rushing to appease it on a particular hay sheaf. No one will be able to foresee in which direction the animal will head, but it is certain that it will feed on any of the sheaves it can find. In doing so, the satiated donkey-hay configuration has lost its initial symmetry.[3]
R.2 - d) Discussion, lines 734-736, “If all physical quantities would be simultaneously measurable, one would soon realize the existence of a “state variable” … However, if there are quantities that are not simultaneously measurable, then a state variable is illusionary.” This statement has strictly no sense in quantum theory. State is defined in quantum theory (both in QM and QFT) by “state variables” which correspond to a “complete set of observables”. Complete set of observables is a maximal set of pairwise commuting observables (i.e., corresponding operators). Quantum state described by a complete set of observables provides a maximal available information on the system. In this respect state variable is not illusory – quantum state is defined by the values of its state variables.
Authors: agree, we reformulated the statements in the paragraph.
R.2 - e) Line 738-739, sentence “The absence of state variable highlights an essential uncertainty and implies that at least two quantities are not simultaneously observable.” should read “The existence of complementary observables highlights …”.
Authors: we implemented your suggestion (line 739)
Hoping that all issues have been addressed properly, we thank you for your constructive feedback.
Sincerely,
Dr. Pierre Madl
Dr. Paolo Renati
[1] Ho MW (1997) Quantum coherence and conscious experience. Kybernetes, 26(3), 265–276. doi:10.1108/03684929710163164
Ho MW (2017) Meaning of Life and the Universe. World Scientific. Singapore. ISBN 978-981-3108-86-8. doi: 10.1142/10012
Ho MW (2016) Science in Society: https://www.i-sis.org.uk/QuantumJazz.php
[2] Pitch A (2018) Effective field theory with nambu–goldstone modes. Effective Field Theory in Particle Physics and Cosmology, 1–70. doi: 10.1093/oso/9780198855743.003.0003 https://arxiv.org/pdf/1804.05664.pdf
[3] Preparata G (2002) Dai Quark ai Cristalli – breve storia di un lungo viaggo dentro la materia (EN: From Quarks to Crystals - brief history of a long journey inside matter). Bollati Boringhieri, Torino (ITA) ISBN: 88-339-1392-9

Round 2
Reviewer 2 Report
I found the present version of the paper satisfactory. The authors have effectively incorporated my comments in the revised version, and so I believe that the manuscript is now ready for publication.